# A Systematic Literature Review of Sustainable Consumer Behaviours in the Context of Industry 4.0 (I4.0)

Ayten Nahide Korkmaz [1] and Meral Uzunöz Altan [2,*]

1 Department of Foreign Trade, Istanbul Aydin University, İnönü Street, No:38, Beşyol, 34295 Istanbul, Turkey; aytenkorkmaz@aydin.edu.tr

2 Faculty of Economics and Business Administration Sciences, Yıldız Technical University, Davutpaşa Campus, Esenler, 34220 Istanbul, Turkey

* Correspondence: muzunoz@yildiz.edu.tr

**Abstract:** Sustainability and related issues are widely accepted as vital themes in contemporary fields. These include the idea of developing products and services necessary for individuals to lead sustainable lives into the future in Industry 4.0 (I4.0), the main focus of which is digitalization. Also, the notion of Sustainable Consumption (SC) is related to the Sustainability Development Goals (SDGs), which comprise SC and production motives. The methodology of this study involved analysing data from a bibliometric review, which was obtained from different themes within environmentally friendly and ecofriendly consumption sources. We searched a key theme, SC, in the Web of Science (WoS) database and obtained 1451 documents. A total of 1005 documents were selected. In the next stage, we searched for two key words, "environmentally consumer behaviour" and "eco-friendly consumer behaviour"; 258 studies were obtained from the WoS database. This paper represents a thorough literature review of the line between consumers and SC. The aim of this study is to underline the level of awareness of environmentally friendly and ecofriendly consumption in the I4.0 period by selecting and examining articles published over the past 10 years. The final aim of this work is to provide suggestions based on gaps in the literature.

**Keywords:** consumer behaviours; eco-friendly; environment; environmental approaches; sustainability; sustainable consumption; Industry 4.0 (I4.0)

## 1. Introduction

Global powers in the economic scene have contributed to many significant effects in nature. The unlimited use of natural resources, disasters, pandemics, industrialization, and urbanization may be factors that make human life chaotic. In addition to this, the uncontrollable and rapidly growing population has brought unlimited consumption habits with it. At present, while individuals make decisions about their consumption under the influence of many factors, such as global warming, energy problems, scarcity, air pollution, and economic crises, they are also affected by the abundance of goods and service options, brands, advertising, income levels, habits, etc. [1]. In order to make life sustainable, conscious consumption behaviours should become a main focus.

Whereas individuals only focused on increasing their living standards before the 1970s, it is thought that the concept of "*sustainability*"—which stood out in all areas, including the environment, economy, and social issues—entered the literature in 1987 after a report titled "Limits to Growth" was published by the Club of Rome in 1972 with the acceleration of production in the globalization process. The concept of sustainability, which was explained by the World Commission of Environment and Development (WCED) as "the way of consumers to have and solve their consumption problems without limiting the amount and of resources that can meet the wants of future comers", was given official importance by the World Commission of Environment and Development (WCED) in 1987. In other

words, we can say that sustainability, whose Latin root "*subtenir*" means "*to protect*" or "*to support from below*" [2], translates to efforts to ensure that the quality and existence of existing resources are preserved and transferred to future generations [3]. According to Gilman (1992) [4], sustainability is defined as the flow of the functioning of a society, ecosystem, or any current system into an unknown future without depleting the main resources; according to Ruckelshaus (1989) [5], it can be defined as "the idea that economic growth and development within the strict lines of ecology will be achieved through interaction and protected over time". It is estimated that, in order to support SC and use resources effectively, the ultimate milestone of changing existing consumption habits towards ones based on sustainability will be achieved through understanding and clarifying the effects of consumer behaviour, as well as by offering solutions.

With the new world order, we see that new phenomena related to sustainability have begun to emerge in line with neo-liberal policies. When the phenomenon of sustainable consumption is analysed, a dual concept is encountered. While examining the concept, it is thought that it is sometimes used alone [6], or it is sometimes used together with the concept of sustainable production [7]. This dichotomy causes the concept of production and consumption to come together and become an inseparable whole. Sustainable consumption does not refer to a reduction in production; rather, it generally emphasizes factors such as consumption avoidance, awareness, environmental awareness, and orientation towards environmentally friendly products. When the literature is examined, the concept of sustainable consumption is shown to be mostly related to environmental sensitivity. However, environmental awareness is only one dimension of sustainable consumption. Behaviours such as replacing products with new ones when new ones are not needed, the understanding of savings during and after the use of products, and the long-term use of the purchased product should also be considered within the scope of sustainable consumption. Based on this idea, it is estimated that sustainable consumption behaviour also includes environmentally sensitive purchasing, although some of the sustainable consumption and sustainable consumption behaviours indirectly affect environmental awareness. It is believed that the sustainability of consumption is ensured by believing that natural resources should be processed for production and included in consumption, and that the needs produced after consumption should not harm either the environment or the resources to be used for production [8]. In order to ensure this situation, the environmentally friendly aspects of sustainable consumption behaviours should be identified. Consumer behaviour is a vital indicator for measuring sustainability in green consumption [9]. According to Muntinga et al. (2011) [10], there are some types of consumer behaviour that consist of consumption and contribution. Eco-friendly behaviour is defined as reactions that aim to protect the environment and humankind [11]. Additionally, the definition of environmentally conscious behaviour includes the idea of reactions necessitating limited costs in order to achieve the goals of environmental sustainability [12]. There are many studies centred on eco-friendly and green consumption behaviour in the literature.

In this research, we aimed to orderly and systematically group the articles selected from the scholars in the WoS database. Then, we aimed to show the positive effect of sustainable consumption and examine consumers' eco-friendly and ecologically friendly behaviours in the context of the development of the I4.0 period. Our other aim was to demonstrate the gap in the literature that is present between sustainable consumption and consumer behaviours by emphasizing the importance of developing the I4.0 period in order to present sustainable, consumption-oriented, consumer behaviour. Its major effects for sustainable life and consumption can be largely made possible via conscious consumption behaviours. Within I4.0, there are many technological offers and ideas for protecting the environment. It is thought that it is necessary to examine "consumer behaviours" and "eco-friendly consumer behaviours" in order to ensure sustainable consumption and, if we evaluate it as a whole, sustainable life. In this paper, our main aim and research focus is to conduct a literature review, grouping the articles in an orderly and systematically manner

by examining consumers' eco-friendly and environmentally friendly behaviours in the context of the development 4.0 concept.

## 2. A Literature Review of Enviromentally Conscious Consumer Behaviour

At present, consciousness in behaviours concerning the environment is a significant theme in the literature (Figure 1). Because of the role of consumers' reactions in changing the environment, it is meaningful to search and assess the intent of consumer behaviours. Environmental consumer behaviour (ECB) is discussed using terms such as the green economy [13]. Environmentally conscious consumer behaviour is defined as green consumption activities which reduce possible negative effects on the environment [14]. According to Pieters, environmental consumption is a combination of actions which have a more positive impact on nature than other actions [15]. Because of their intent of improving the environment, people try to change their behaviours [16] and, in the environmental marketing literature, ECB is a special field defined as a set of intents and thoughts about goods and products concerning the environment [17]. According to Albayrak et al. (2013) [18], environmentally conscious consumers can be identified through three aspects, which include many sides, and in our study findings, we understand that consumers who are sensible to the need to protect nature intend to behave in an environmentally friendly manner.

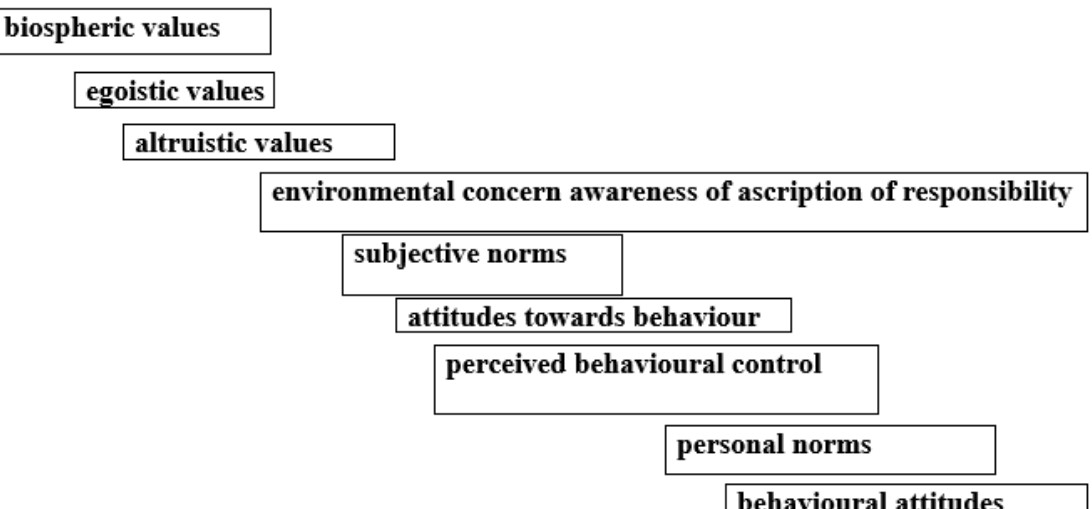

**Figure 1.** The variables that affect the environmentally conscious consumer behaviour. Source: Designed by the authors and based on [19].

## 3. A Literature Review of Eco-Friendly Consumer Behaviour

The notion "eco", which is used as a short form of the word "ecology", examines the relations between organisms and their ecological place. Because of this, "eco"-friendly (or "ecology-friendly") is defined as actions which have the aim of diminishing negative effects on nature. At present, eco-friendly goods and services are popular in the marketplace and producers have their sights on the production of eco-friendly outputs. Eco-friendly products are defined as products which are renewable, can be recycled and preserved, and do not pollute the world [20]. Eco-friendly products have some vital qualities [21]: (1) they do not have negative effects on living beings; (2) they protect nature; (3) they use limited resources in their disposal; (4) they are renewable and can be recycled; (5) they contain materials which do not harm the environment. According to Mosiander, ecofriendly consumer behaviour refers to green consumption and the aim of protecting the environment [21], which are related to environmentally friendly, green or sustainable reactions [14]. At present, people are focused on the consumption of eco-friendly products [22] In addition to this, because of the limited resources and regulations, lots of producers and sellers are aware of eco-friendly consumption [23]. According to the data from "2018 Green Sustainable Consumption Promotion Week", in China, the number of people who are sensible of the

need for eco-friendly consumption rose from 58% to 83% [24]. To support this idea, some regulations are seen; in Vietnam, plastic products have been forbidden since May 2020 and, at present, the United States, South Korea and Taiwan have introduced a restriction on the use of nylon bags [25]. It is understood that eco-friendly products and consumption are vital in human life [26].

## 4. Materials and Methods

This systematic review aims to compile articles about sustainable consumer behaviours in context of Industry 4.0. In order to search the studies, the key concepts "environmentally consumer behaviours" and "eco-friendly consumer behaviours" were the keywords searched in the Web of Science (WoS) and Scopus databases. The study concerned mentions of sustainable consumption (SC) and "environmentally" and "eco-friendly" behaviour in the literature. In this way, researchers had the chance to read articles about the vital notion of sustainable consumption (SC) and these two concepts. We selected articles according to the the principles seen in the flow chart (Figure 2). In the first step (1), we determined the key words (environmentally consumer behaviours and eco-friendly consumer behaviours); in the second step (2), we examined the Web of Science and Scopus databases by searching the keywords; in the third step (3), we compiled the articles; in the fourth step (4), we selected the articles and assisted in the search.

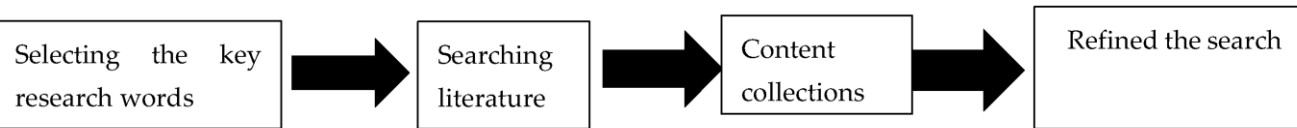

**Figure 2.** The research method process path. Source: designed by the authors.

We used a Systematic Literature Review (SLR) (Figure 3) with four criteria. Firstly, we carried out a systematic literature review, starting with planning [27]. Secondly, during the selection step, the articles were searched according to the keywords "environmentally consumer behaviours and ecofriendly consumer behaviours". Then, articles were obtained and evaluated. The final criteria are examining the articles and writing the review [27].

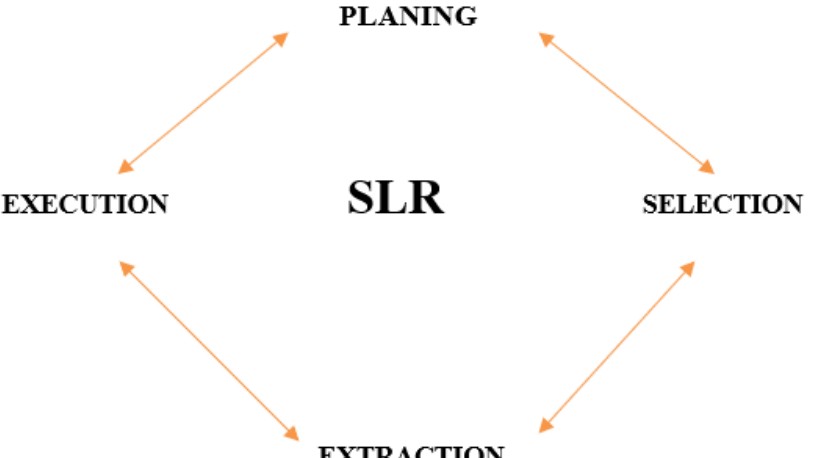

**Figure 3.** Basic structure of a systematic literature review. Source: designed by the authors and based on [27].

The Systematic Literature Review (SLR) method is known as a qualitative analysis which contains assessment and examination steps and outlines the seven key principles behind systematic literature reviews [28], as implied by scholars and researchers [29]. This method has seven vital qualities: transparency, clarity, integration, focus, equality, accessibility and coverage. Search, Appraisal, Syntheisis and Analysis (SALSA) is very detailed other method that contains SLR [30]. SLR is an also academic method that is

not related to the others and aims to define and evaluate the literature that is relavent to the chosen topic. In the final stage of this method, conclusions about the key concept are detailed systematically. This methodology guarantees clarity, flowness and vertility, because it is an approach that aims to diminish the risks and obtain a principled definition [29].

### 5. Analysis and Results

The literature search aims to assess only the articles that present environmentally friendly and eco-friendly concepts against an economic background. Almost all the studies were searched using the behavioral concept and included other disciplines. However, as the economy is a multi-disciplinary field, we tried to compile the studies which solely focused on the economic field. The need to search a multi-disciplinary field like the economy presents a huge obstacle to researchers (Figure 4 and Table 1).

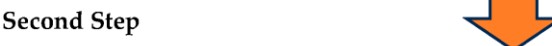

**First Step**

We searched a key theme "Sustainable Consumption" in the Web of Science (WoS) database and reached 1451 documents

**Second Step**

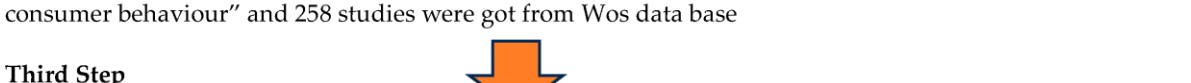

Subsequently, after examining the studies aimed to get only articles and review articles. 1005 documents were selected from all. In the next stage, the two key words "enviromentally consumer behavior" and "eco-friendly consumer behaviour" and 258 studies were got from Wos data base

**Third Step**

In 280 studies, "the relationship between tax and income inequality" were analyzed only superficially as they approached it indirectly.

**Figure 4.** Summary of the procedure used to obtain the article sample. Source: designed by authors and based on [9].

**Table 1.** Definitions of SC in the literature. Source: designed by authors.

| Sustainable Consumption (SC) | |
|---|---|
| **References** | **Definitions** |
| United Nations Conference on Environment and Development, 1992 [31]. | SC is the responsibility of individuals to sustain environmental processes and diminish effects which harm nature. |
| Gabriel and Lang, 1995 [32]. | From a traditional perspective, consumption is a measure of a society's continued development and prosperity. |
| OECD (Organization for Economic Cooperation and Development, 2002 [33]. | SC defined as meeting basic needs and ensuring quality of life through the consumption of goods and services without affecting the needs of future generations. Although this definition can be interpreted in different ways, there is a consensus among developed countries that the use of resources should be reduced. |
| Harron et al., 2005 [34]. | SC is key to ensuring that a high standard of living can be maintained. |
| Barber, 2007 [35]. | SC indicates environmental needs and the motivation to suit the needs of individuals from different situations all over the world. |
| UNEP, 2011 [36]. | SC aims to maintain the welfare of the environment and the needs of the next generation. |
| Barroso et al., 2011 [37]. | SC tries to support new ideas and improve their sustainability. |
| De Camillis et al., 2013 [38]. | SC has diminishes the effects on human life. |
| Alisat&Reimer, 2015 [39]; Bogueva et al., 2017 [40]. | SC limits the superfluous consumption of natural resources. |
| Chekima et al., 2016 [41]. | SC is an indicator of sustainable development occurring far from policymakers. |
| Farr, 2018 [42]. | SC supports actions aiming to sustain future generations' good quality of life. |
| Böhme et al., 2018 [43]. | SC is a method which paves the way for both cognitive and emotional realities. |
| Luthra et al., 2020 [44]. | SC relates to the cultural motives and obtains information from these patterns. |
| Mochis et al., 2021 [45]. | SC aims to shape food-based needs so that they do not lead to sustainable bad habits. |

## 6. Theoretical Framework of Sustainable Consumption in 4.0 Period

Ecological citizenship, on the other hand, is a phenomenon that enables us to predict how we can reduce the ecological impact of the daily behaviors of all living people and questions how we should live [3]. According to the Organization for Economic Cooperation and Development (OECD) "sustainable development (SD)" is "the new ways to deplete of goods and services aimed to get better quality of life conditions while controlling the limit of the resources like water, energy and finding new methods to control this resources and sustain them to the future generations". The notion Green Economy (GE) has been defined as "good conditions in life while there are huge environmental risks and scarcity in ecological resources" [36].

## 7. Studies on SC Related to EFC and ECB in the Context of the I4.0 Period

The articles shown in Table 2 are related to ECB and those presented in Table 3 are related to EFB. This section aims to obtain a detailed systematic representation of the studies.

**Table 2.** The studies on ECB in the I4.0 period. Source: designed by authors.

| Author(s) | Method(s) | Country(s) | Conclusion |
|---|---|---|---|
| Chockalingam and Isreal (2016) [46] | Survey | India | The research sheds light on the suggestions and new solutions aiming to obtain eco-friendly products and their components. |
| Verain et al. (2016) [47] | Data Collection | Belgium | The findings show that the more new solutions there are involving sustainable products, the greater the number of successful paths that can be followed. |
| Geng et al. (2017) [48] | Data Collection | China | The findings show statistically significant effects in developed countries that have made more efforts to make adolescents aware of SC behaviours. |
| Tasci (2017) [49] | Descriptives, frequencies, *t*-test and one-way ANOVA test using SPSS version 24. | USA | The results show that people give priority to short-term advantages and that gender is a key factor explaining sustainability behavior. |
| Ritter et al. (2017) [50] | Data Collection and Analysis | Brazilia | This study pays the way for the introduction of sustainability in consumption in southern Brazil. |
| Nam (2017) [51] | Data Collection and Analysis | Korea | This study presents details of consumer concerns. |
| Zver and Vukasović (2020) [52] | Data collection with questionnaires | Slokvia | The results of the research were for planning for sustainable marketing concerns. |
| Mochis et al. (2020) [45] | LCP method | Asain countries | In sum, this article shows that LPC is a vital method for assesting sustainability. |
| Lee and Chang (2022) [53] | Data Analysis | China | This paper aims to present the findings regarding China's sustainable consumption habits, and, according to the study, these behaviors are improving. |

**Table 3.** The studies on EFB in the I4.0 Period. Source: designed by authors.

| Author(s) | Method(s) | Country | Conclusion |
|---|---|---|---|
| Behe et al. (2010) [54] | Data Collection | Indiana, Michigan, Minnesota, and Texas. | This study claims that ecobehaviours should be more effective to lead to action retgarding herbal and plant consumption. |
| Nhamo (2010) [55] | Literature Review | South Africa | This study seeks to induce a transition to a green economy. The new jobs are related to the green jobs and planned policy. This study aims to close the gap in the literature. |
| Jang et al. (2010) [56] | Data Collection Analysis | South Korea | In this study, many scholars examined the vital behavioral differences regarding the green economy between two cultures in Korea. |
| Okada and Mais (2010) [57] | DataAnalysis | Hawai | The results show that the advantages of the consumption of green products are shown to be increasing at present, and offers new ways to induce these actions. |
| Tseng et al. (2013) [58] | Literature Review | Vietnamese | According to this study, resources should be used in an efficient way to be sustainable. |

**Table 3.** *Cont.*

| Author(s) | Method(s) | Country | Conclusion |
| --- | --- | --- | --- |
| Engilis and Philips (2013) [59] | Literature Review | United States | This study produces new environmentally friendly products and offers for consumers |
| Paul et al. (2015) [60] | CEF and SEM | Norway | This study will develop new ways for policymakers and scholars to centre green consumption. |
| Loiseau et al. (2016) [61] | Literature Review | UN members | This study examines the concept of the green economy and related concepts and approaches, and aims to make a contribution to the literature and establish a pathway to a green economy. |
| Yadav and Patak (2016) [22] | Data collection and SAM | Greece and UK | The study presents some offers to young consumers regarding the green consumption, suggesting that these have some limitations that should be addressed in further research. |
| Korhonen et al. (2017) [62] | Literature Review | China | The paper presents a new piece of conceptual research about CE. |
| Gunden et al. (2019) [63] | Literature Review | Turkey | This study aims to discover consumers' eating habits and green consumption. |
| Kautish et al. (2019) [64] | AMOS model | European countries | This study is the first study representing the TRA/TPB framework to obtain data on environmental consciousness in this context. |
| Stukalo and Sımakhova (2019) [65] | Literature Review | Literature Review | This study's analysis suggests that it is possible to make key proposals to strengthen the environment. |
| Golob and Kronegger (2019) [66] | Modelling approach | 28 EU members | The aim of this study was to find new solutions improve environmental consciousness and segment European Union (EU) consumers using the structural method to support the idea of SC. |
| Novska (2019) [67] | Therotical Model | Macedonia, Albania, Croatia, Bosnia and Herzegovina, Israel, Czech Republic, Hungary, Switzerland and United States | This study aims focuses on economic policies, aiming to create new ways to make them more effective. |
| Khojasteh-Khosro et al. (2020) [68] | ANP and ANOVA Method | Switzerland | According to the study, consumers should priotize products which are made of green materials. |
| Macaulay (2020) [69] | Data Collection | Africa | According to the paper, policymakers should take recycling as a first solution, as well as offering new solutions. |
| Reijonen (2020) [8] | Literature Review | - | This paper offers a new booklet focusing on socio-material factors and offers new policy actions for green consumers. |
| Kristoffersen et al. (2020) [70] | Literature Review | - | This research aimed to map the other research. It collected other studies about CE and organized articles related CE. |

**Table 3.** *Cont.*

| Author(s) | Method(s) | Country | Conclusion |
|---|---|---|---|
| Mainardas et al. (2020) [71] | Structural Model | Brazilian | In this paper, new green behaviours are examined and the effects of the green consumption are presented. |
| Moon (2021) [72] | Data Analysis | Korea | New solutions related to the consumption of green products were recommended in this paper |
| Preut et al. (2021) [73] | Literature Review | Canadian | This article outlined several areas of interest regarding the social economy and circular economy. In general, the social economy supports the circular economy in important ways. The main point revealed in this general introduction is that the social economy challenges us to think about circular economy in a broader sense: not only in terms of who has a say but also in terms of who can participate and who is included. |
| Lakatos et al. (2021) [74] | Systematic Review | China&Japan | According to the paper, cities can induce a circular economy; the paper tries to show the big picture to scholars. |
| Cheng et al. (2022) [26] | Data analysis | Austria | This article suggests subjective ways to control the behaviors of consumers. |
| Dewick et al. (2022) [75] | Literature Review | China | The results aim to attract attention from scholars by presenting detailed knowledge about CE. |
| Çoker and Linden (2022) [76] | HLRSM Model | United Kingdom | The aim of this study is to analyze the effectiveness of the TPB and its results regarding green economy consumer behaviours. |
| Radulascu et al. (2022) [77] | Literature Review | - | This study pays attention to the unique qualities of each region and aims to achieve global effects. |
| Mealy and Teytelboym (2022) [78] | GCI(HH) and GCI(Tacch) | UN members | The study results shed new light on the green economy and its effects on the industry. |
| Oliver et al. (2023) [79] | Data Collection | US | The study's results represent new ideas about green consumption and eco-friendly behaviour regarding green products and aims to attract the attention of environmentally green consumers. |
| Vlastelica (2023) [80] | EFA& SEM | China | According to this research, young educated consumers seemed to be aware of green consumption. |

## 8. Classification of the Articles

The selected articles are defined according to particular concepts: "by countries", "by the themes which are mostly underlined", "by the themes which are mostly underlined", "by the methods of the articles", "by number of the year publication" and "by offers". This section aims to pay the way for scholars and researchers in the future.

### 8.1. By Country

The articles in this systematic review mainly focus on m Albania, Belgium, Bosnia, Brezilia, Canada, Czech Republic, China, Croatia, Greece, Hawaii, Herzegovina, India, Israel, Korea, Macedonia, Michigan, Minnesota, Slovakia, United Kingdom, Hungary, Switzerland, Turkey, United States, and Vietnam. However, three of the studies deal with ECB and EFB in UN member countries, but do not offers more information on the included countries. For example, some researchers conducted their studies in the same country; Chockalingam and Isreal (2016) [46] conducted their study in Indiana and Behe et al. (2010) [54] obtained their data from analyses in Indiana, Michigan, Minnesota, and Texas. South Africa is the country in which Nhamo (2010) [55] and Jang et al. (2010) [56] did their research. Geng et al. (2017) [48] and Lee and Chang (2022) [26] wrote articles about ECB in China; similarly, Dewick et al. (2022) [75], Korhonen et al. (2017) [62] and Vlastelica (2023) [80] studied EFB in China.

### 8.2. By the Themes Which Are Most Underlined

The notions of SC, EC, ECF and ECB are discussed in several studies focusing on the green economy and sustainable green consumption. Most of the relevant studies are related to sustainable products, green economic solutions, the circular economy, and agriculture, which is covered in 14 studies. The circular economy is addressed in three studies, as well as marketing strategies for SC, and offers to provide education on ECF and ECB and CE. Ritter et al. (2017) [50], Nam (2017) [51], Korhonen et al. (2017) [62], Lakatos et al. (2021) [74] and Vlastelica (2023) [80] present new ways to ensure the sustainability of CE.

### 8.3. By the Methods

Many different methods were used in the articles which were selected in this research. Bibliometric analysis was common, as well as literature reviews. Most data-collection methods comprised surveys and detailed questionnaires (Chockalingam and Israel 2016) [46]. Novkovska (2017) [67] used a theoretical model that has previously been used to determine the efficiency of the sustainable economy. Mealy and Teytelboym (2022) [78] used a theoretical situation for their analysis. In addition this, EFA, SEM, CEF, GCI(HH) and GCI (Tacch), as well as LCP and HLRSM, were conducted on the articles which were selected. For example, Okada and Mais (2010) [57], Jeng et al. (2020) [56], Yadav and Patak (2016) [22], Nam (2017) [51], Verian et al. (2017) [47], Ritter et al. (2017) [50], Çoker and Linden (2020) [76], Zver and Vukasović (2020) [52], Macaulay (2020) [69], Moon (2021) [72], Lee and Chang (2022) [53] and Oliver et al. (2023) [79] used data collection and an HLR model.

### 8.4. By the Year of Publication

The articles included in the review date from 2019 to 2023. The number of articles per year focusing on ECF and ECB has risen steadily, as shown in Figure 5. From the year 2021 onwards, the number of publications increased, reaching the top level in 2023.

Some articles are studies in which the scholars try to understand the adoption of SC and EC, as well as their implications. According to Novskova (2019) [67], the vital contribution of the study regard the motivations of the hidden economy, in which green economy is a complex link between the hidden economic segmentations, flowing from sector to sector and serving as a conductor that can sustain GC and SC.

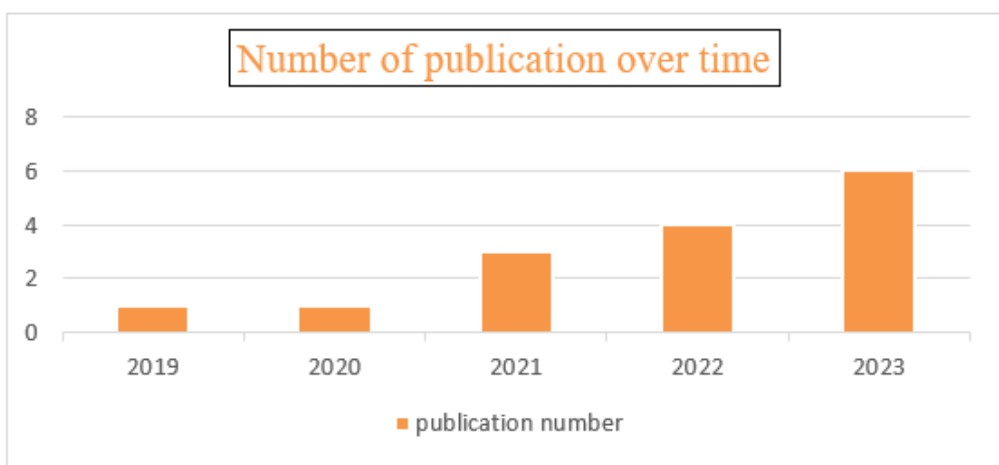

**Figure 5.** Articles published per year on SC. Source: based on authors.

*8.5. By Offers*

Several articles which are examined in this study presented offers to consumers, organizations and policymakers. For example, according to Verain et al. (2016) [47], "eco-friendly" and "ecologically friendly" consumer behaviours should be oriented toward sustainable food consumption [47]. In addition to this, according to Geng et al. (2017) [37], the key component in the use of sustainable products is education. This article implies that the precondition for purchasing green products is understanding the importance of green sustainability [37]. Nhamo (2010) [55] has suggested that EC and ECF and ECB may provide chances to present new opportunities regarding sustainable jobs tailored toward the green economy. The conclusions of some articles show that most implications should be relatedto companies or policymakers. For example, in one study, the presentation of the most useful applications for each stage of industry will be a top priority in sustainable production [75]. According to another study, ideas about green operations to manage SC might be analyzed step by step [81]. Another study [62] aims to present the strategies and goals of CE. In this study, the authors want to underline the specifications of the literature and address the definitions of the concept, as well as presenting the methodological models and steps used to show the challenges of the notion. Tseng et al. (2013) [58] claim that SC is a complex notion that can provide new approaches in industries and will continue to function in the future. Garnetet al., 2019 [82], claim that, in the future, the findings of the study could be expanded to present objective standards for behaviours, the study could be repeated in different cultures, and the moderating role of individuals' awareness of the depletion caused by meat consumption and global warming could be researched. Hall et al. (2010) [83] showed that consumers made different choices about sustainable products but the vital aspect was that their decisions seemed to be made based on the factors of price, footprint, etc. According to Okada and Mais (2010) [57], the consumption of green products increases because of actions taken to increase awareness of ECF and ECB.

In summary, in this systematic review, we aimed to define the major enablers or effects of green marketing styles, as well as consumer behaviours in the IE4 period. While searching these articles, we obtained vital results. First of all, we assessed a large number of studies, as represented in Tables 2 and 3, that took place over 10 years, stored in the WoS database. During the selecting and compiling process, the key components became helpful and we saw that there has been a major advancement in the academic research in last 10 years, which should continue in the future. According to this research, new solutions should be investigated and presented to policymakers. Academy and policymakers should act together to ensure SC, GE and ECB and EFB. This study is thought to serve as a map which compiles and organizes articles related to "eco-friendly" and "ecologically friendly" consumer behaviours in the context of I4.0.

## 9. Study Limitations and Directions for Future Research

This study has several limitations which offer additional avenues for future research. This study is based on a sample drawn from the WoS database. Further, within the WoS database, we drew our sample from "Eco-friendly" consumer behaviour, "Ecologically" consumer behaviour and "Industry 4.0" subject areas. Future researchers could draw their samples from other databases, such as Scopus and Wiley (Table 4).

**Table 4.** Future research directions. Source: based on authors.

| Category | Future Areas of Research |
|---|---|
| SC(1) | SC is known as a vital component of development. (Zheng et al., 2021) [84] |
| SC(2) | SC may have been seen as an instrument for "ecological citizenship", which is defined as a kind of citizenship that individuals can achieve through their political and environmental choices in their private consumption behaviour. (Seyfang, 2006) [3] |
| ECB(1) | In the literature, a huge number of explanations about ECB can be seen; the concept can be described as the attitudes and thoughts about green environmental consumption. (Fontes et al., 2021) [85] |
| ECB(2) | Consumers use eco-friendly products to protect new generations from the harms caused by negative effects on nature. The green economy is made up of ecofriendly consumer behaviour (ECB), which is an important motive with lots of positive side effects caused by the utilization of efficiency to ensure sustainability. (Jarayaraman et al., 2019) [86] |
| I4.0(1) | The notion of Industry 4.0 is rooted in a project initiated by the German government's high-tech strategy to promote the computerization of production. (Song, 2017) [87] |
| I4.0(2) | The digitalization period started with computers, became widespread thanks to the internet, and continues with Industry 4.0, which is the ability of machines, computers and people to exchange data with each other via the internet (Ozdoğan, 2017) [88]. |
| I4.0(3) | I4.0, whose emergence is a technology-centered economic policy and dates back to the 2011 Hannover Fair in Germany, refers to digital transformation processes that not only include human–human and human–machine interactions, but also paves the way for machines to interact with each other (Rojko, 2017) [89]. |
| I4.0(4) | A system that is restorative and regenerative by design, which aims to maintain products, components and materials at their highest level of utility and value (Spaltini et al., 2021) [90]. |

## 10. Conclusions

Due to the increasing use of natural resources, environmental problems have increased rapidly. At present, both governmental and non-governmental organizations and consumers have started to think about new solutions to protect nature and take some new steps to solve environmental problems. The concept of sustainable consumption in the 21st century, which is based on the understanding of the need to transform consumption, has gradually begun to replace the previously endless desires and consumption habits to guarantee the quality of life of both present and future populations, and sustainability practices have become important for scholars, organizations, and policymakers in recent years [91]. It can be observed that environmental problems are associated with production rather than consumption. However, in the last 25 years, the rapidly increasing environmental disasters and the social, ecological and economic deformations accelerated by overconsumption have shifted the main focus of attention from production to consumption. The notion of sustainable consumption is seen as a new solution.

Consumers have started to be conscious of products that have some different qualities, which are called "eco-friendly" and "environmentally friendly". The production of these goods and services has significantly improved and consumers' awareness of the importance of recycling materials has strongly increased. Because of the Fourth Industrial Revolution Era, the strategy of mass personalization has made some changes to production. Reactions will have to change from the root to the product to network services. This can provide many new ideas for the production of unknown concepts. Original ideas that can be manufactured using sophisticated networks of enterprises using Industry 4.0 technologies means the creation of fresh and new ways to communicate with consumers. Furthermore,

the amount of product creation will be determined according to the aim of creating only a physical product that will lead to new experiences and consumers' wants. Because of this, the implicationso of I4.0 seem to be closely related with ECB, EFB and SC.

In more detail, this study can serve as a map for scholars aiming to analyze articles about SC and eco-friendly and environmentally friendly behaviours in the context of I4.0. These studies may contain some fruitful suggestions. SC and the green economy might come to the fore because, in addition to the win–win decisions of policymakers, there will be new job opportunities, which will foster the well-being of human beings globally.

In this paper, the literature was searched using two key concepts in the WoS database. This paper reviewed the literature, aiming to open a new pathway to ensure a pure combination of these two key criteria. This paper provides a thorough literature review of the relationship between consumer awareness and sustainable consumption. The methodology of this study is that the data were analysed by a bibliometric review study, which focused on different themes regarding awareness of ecofriendly consumption. To achieve this aim, a systematic literature review was carried out, selecting articles published over the past 10 years included in the Web of Science database. To help to overcome these challenges, we used two key features, "environmentally consumer behaviour" and "ecofriendly consumer behaviour", to categorize the papers. We estimated that this would provide a good starting point to understand the factors that determine consumer behaviour in order to ensure sustainable consumption and to use resources effectively, to change the existing consumption habits to improve sustainability, and to offer solutions.

**Author Contributions:** Conceptualization, A.N.K. and M.U.A.; methodology, A.N.K.; software, M.U.A.; validation, M.U.A. and A.N.K.; formal analysis, M.U.A.; investigation, A.N.K.; resources, M.U.A.; data curation, A.N.K.; writing—original draft preparation, A.N.K.; writing—review and editing, M.U.A.; visualization, A.N.K.; supervision, M.U.A.; project administration, M.U.A.; funding acquisition, A.N.K. All authors have read and agreed to the published version of the manuscript.

**Funding:** This research received no external funding.

**Institutional Review Board Statement:** Not applicable.

**Informed Consent Statement:** Not applicable.

**Data Availability Statement:** The data that support the findings of this study are openly available in the web of science at https://www.webofscience.com/wos/woscc/basic-search (accessed on 17 August 2022).

**Conflicts of Interest:** The study does not present any conflicts of interest for the authors or other stakeholders.

## Abbreviations

| | |
|---|---|
| CE | Circular Economy |
| ECB | Environmental Consumer Behaviour |
| EFA | Exploratory Factor Analysis |
| EFC | Eco-friendly Consumer Behaviour |
| EFP | Eco-friendly Product |
| GE | Green Economy |
| GC | Green Consumption |
| I4.0 | Industry 4.0 |
| OECD | Organisation for Economic Co-operation and Development |
| SALSA | Search, Appraisal, Synthesis and Analysis |
| SC | Sustainable Consumption |
| SD | Sustainable Development |
| SDG | Sustainable Development Goals |
| SL | Sustainable Life |
| SLR | Systematic Literature Review |
| WCED | World Commission of Environment and Development |
| WoS | Web of Science |

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
