# Peer review of "A Systematic Literature Review of Sustainable Consumer Behaviours in the Context of Industry 4.0 (I4.0)"

_sustainability, doi:10.3390/su16010126_

Round 1

Reviewer 1 Report

Comments and Suggestions for Authors

Dear authors,

Firstly, I would like to thank you for your work in this field.

You did a good job!

After extensive reading, I only have a few comments for the current version, which are not many but essential.

For revision details, please see the following:

[1] The title is a bit long, I suggest shortening it.

[2] I suggest you add more keywords and sort the words alphabetically. It can help you improve the manuscript's exposure.

[3] I suggest you re-structure the abstract with the following steps: Research background + Research intention + Methodology & Data research + Specific research content + Main finding + Conclusion from the main finding + Research meaning, Applying experience, etc.

[4] I suggest you double-check the journal format, the current format does not match it.

[5] the deep of the introduction is not enough. I suggest you re-order it with the following structure: Development background + research background + research review + research aim + paper overview.

[6] I suggest you re-draw the figure 2, I don't really understand its internal logic/meaning/value.

[7] chapters 2 and 3, you presented a lot of literature, I suggest you point out/emphasise the analysis/review of them, not just present.

[8] you presented a lot of information in this manuscript, I suggest you add a chart/flow/figure to show the whole technique routine. it can help readers understand what did you do rapidly and help them to understand the core value of this research.

[9] I suggest you reorder the conclusions chapter with the following structure: research issue + research result + research value + research direction in the future.

If my comments are properly considered and followed, I have no further comments on this manuscript.

Again, I have a high comment on your current contribution. All my upon-revision recommendations are intended to help this manuscript be more readable for readers.

Comments on the Quality of English Language

minor check

Author Response

Dear Reviewer,

Thank you for your insightful feedbacks. We appreciate your suggestions and have acted upon them. These additions will empower readers to better comprehend the significance and reliability of our research outcomes.

Best regards,

Reviewer 2 Report

Comments and Suggestions for Authors

Thank you very much for the opportunity to analyze the content submitted to the prestigious Sustainability journal.

First of all, I would like to point out that, if I'm not mistaken, the formatting style of the references is not appropriate. In the body of the text, in the case of Sustainability, you should insert numbers and not the authors' names. Please make sure of this.

The abstract lacks some important aspects for consideration. A good abstract usually has well-defined parts, which are: contextualization, a brief description of the concept, the objective of the study, the method used to carry out the research, the main results and the main contributions. In the case of your work, you left the objective until last, didn't show the main results and didn't show the main contributions either.

In the introduction, the passage "The phenomenon of sustainability thus appears at every stage of environmentally friendly production and consumption. In order to make life sustainable, it is possible to focus on conscious consumption behaviors to a great extent. As can be understood from all these situations, it can be emphasized that to increase awareness of sustainable consumption behaviours. It has become imperative to find new and untested solutions" is too strong for there not to be a corroborating quote.

Although the introduction does a good job of contextualizing the state of the art on environmentally friendly consumption, it lacks some very important aspects. You see, it is essential that a good introduction points to a good research gap. In your penultimate paragraph, you say that there are many studies on environmentally friendly consumption, and this kind of citation is totally at odds with an argument about the importance of your study for the scientific community. What's more, you failed to show studies similar to yours, so that you could highlight a research gap that these studies described had failed to fill. Please see this aspect in the article linked at https://www.cell.com/heliyon/fulltext/S2405-8440(22)01303-2. In this way, you would be able to add value to your work and announce to the scientific community why your work deserves readers' consideration. 

As far as the scientific method is concerned, I think it needs to be better structured too. A high-impact, international publication is expected to have at least a methodological flow, and to be able to present the steps outlined in the research in a didactic way. With this in mind, I'm sending you an article on how bibliometric methods can be structured very well: https://www.mdpi.com/1996-1073/15/3/691. I don't suggest that you cite this work or use the same structure, but find a structure that is suitable for your work and that you can at least structure a figure for your work.

In the results section, you start the presentation with a figure, before you have a text description to mention it. 

You mention that the results correspond to section 5, but each type of result you continue to list. Wouldn't it be better to delete this section "5" and leave the results for each of the sections you've listed? 

In section 7, you inserted a table and made absolutely no comments. This is a serious flaw in the organization of an academic study.

Anyway... in a bibliometric study, authors usually organize the quantitative information first, and then organize the qualitative information. In this article, you're mixing the two patterns and putting the results together almost at random, without a logical sequence and based on discussions. 

In the case of the conclusion, it is important that the authors pay attention to the most crucial points: a reflection on the results of the study, how the work achieved the objectives and answered the research question, the main applied and managerial contributions, limitations and suggestions for future study. Here, the authors review the results of the study, which should not be done in order to produce a good conclusion.

In view of the aspects covered, I believe that the authors should restructure the work and use many other bibliometrics and academic works as benchmarks. Publication in high-impact journals such as Sustainability requires a more consolidated scientific maturity than I had the opportunity to identify in this scientific article. I wish the authors the best of luck and hope that you manage to overcome the necessary barriers to give visibility to your next work.

Comments on the Quality of English Language

No comment

Author Response

(The authors gave the same response as above.)

Reviewer 3 Report

Comments and Suggestions for Authors

The authors conducted a systematic literature review of sustainable consumer behaviours in the context of Industry 4.0. The paper is well written.

1.    Please define research questions and specify where you answered these.
2.    Please be a bit more specific with the methodology you applied for the systematic literature review.
a.    What is the exact keyword you used?
b.    What are the inclusion and exclusion criteria?
c.    Please specify the process of excluding papers. In figure 4, please add the exact reason for the exclusion of the papers at every step.

Author Response

(The authors gave the same response as above.)

Round 2

Reviewer 1 Report

Comments and Suggestions for Authors

no further comments.

Author Response

Dear Reviewer, we highly appreciate your comments and professional advice, 
which help us improve the quality of our study.

Reviewer 2 Report

Comments and Suggestions for Authors

A brief statement was missing before Chapter 5. The chances improved the Quality of the manuscript. Comgratul

Author Response

Dear Reviewer, thanks to your advice, we had added a brief statement before 
Chapter 5 in revision 1. We are grateful for your valuable contribution, which help us improve the quality of our study.